# Pathogen Exposure in White Whales (*Delphinapterus leucas*) in Svalbard, Norway

**DOI:** 10.3390/pathogens12010058

**Published:** 2022-12-29

**Authors:** Ingebjørg H. Nymo, Ursula Siebert, Christine Baechlein, Alexander Postel, Eva M. Breines, Christian Lydersen, Kit M. Kovacs, Morten Tryland

**Affiliations:** 1Norwegian Veterinary Institute, Holtveien 66, N-9016 Tromsø, Norway; 2Department of Arctic and Marine Biology, UiT The Arctic University of Norway, Hansine Hansens veg 18, N-9019 Tromsø, Norway; 3Institute of Virology, University of Veterinary Medicine Hannover, Buenteweg 17, 30559 Hannover, Germany; 4Lower Saxony State Office for Consumer Protection and Food Safety, Food and Veterinary Institute Braunschweig/Hannover, 30173 Hannover, Germany; 5Norwegian Polar Institute, Framsenteret, Hjalmar Johansens gate 14, N-9296 Tromsø, Norway; 6Department of Forestry and Wildlife Management, Inland Norway University of Applied Sciences, N-2480 Koppang, Norway

**Keywords:** Arctic, beluga, *Brucella*, climate, influenza A virus, morbillivirus, *Toxoplasma*, zoonosis

## Abstract

The Svalbard white whale (*Delphinapterus leucas*) population is one of the smallest in the world, making it particularly vulnerable to challenges such as climate change and pathogens. In this study, serum samples from live captured (2001–2016) white whales from this region were investigated for influenza A virus (IAV) antibodies (Abs) (*n* = 27) and RNA (*n* = 25); morbillivirus (MV) Abs (*n* = 3) and RNA (*n* = 25); *Brucella* spp. Abs; and *Toxoplasma gondii* Abs (*n* = 27). IAV Abs were found in a single adult male that was captured in Van Mijenfjorden in 2001, although no IAV RNA was detected. *Brucella* spp. Abs were found in 59% of the sample group (16/27). All MV and *T. gondii* results were negative. The results show that Svalbard white whales have been exposed to IAV and *Brucella* spp., although evidence of disease is lacking. However, dramatic changes in climate and marine ecosystems are taking place in the Arctic, so surveillance of health parameters, including pathogens, is critical for tracking changes in the status of this vulnerable population.

## 1. Introduction

White whales (*Delphinapterus leucas*) have a circumpolar Arctic and sub-Arctic distribution, with numerous subpopulations using a diverse range of habitats [1]. The global population estimate for the species is around 200,000. In the Svalbard Archipelago, white whales are year-round residents that occupy a very tight coastal zone [2,3]. The species is listed in the International Union for Conservation of Nature (IUCN) Red List as “Least Concern” [4], however, the Svalbard population has been listed as “Endangered” in the Norwegian Red List since 2021 [5]. Aerial surveys that were performed in 2018 estimated the size of the Svalbard population to be 549 (95% CI: 436–723) individuals [6]. This makes the Svalbard white whale population one of the smallest populations in the world [4], which is a concern because small populations are more vulnerable to challenges such as climate change and diseases [7,8].

The occurrence of infectious disease outbreaks in marine mammals has increased globally over recent decades [9,10,11]. This has been linked to climate warming [12], which is taking place particularly rapidly in the Arctic [13]. Viruses have been the main etiological agents identified in marine mammals, specifically morbilliviruses (MVs) and influenza A viruses (IAVs) [12].

IAVs represent a genus within the *Orthomyxoviridae* family. The serological subtypes of their primary viral surface proteins, hemagglutinin (HA) and neuraminidase (NA), are used to classify them [14]. Various subtypes have been detected in cetaceans (H1N3, H13N2, H13N9) and pinnipeds (H7N7, H4N5, H4N6, H3N3, H1N1, H3N8, H10N7, H5N1, H5N8) [15,16,17]. Some of these variants are thought to have caused significant mortality events in seals, such as the H5N1 and H5N8 subtypes, which are found in the ongoing avian pandemic [16,17]. The first documented occurrence of IAV in marine mammals involved the subtype H1N3, which was found in *Balaenopteridae* (species not specified) harvested from the South Pacific in 1975 [18,19]. Sequencing of the nucleoprotein gene showed that the virus was closely related to the one that was circulating in terns (species not specified) [20]. There are two viruses, characterized as H13N2 and H13N9, which have since been isolated from a sick long-finned pilot whale (*Globicephala melas*) “herded ashore” and euthanized on the coast of Maine, USA. The isolates were closely related to those circulating in gulls (species not specified) [21]. IAV Abs have also been detected in several other marine mammal species [15,22,23,24]. The white whale is the only Arctic cetacean, so far, shown to have Abs against IAV (Table 1). Serological investigations of the other two Arctic cetaceans, the bowhead whale (*Balaena mysticetus*) [25] and the narwhal (*Monodon monoceros*) [25], have yielded negative results. IAVs have zoonotic potential. Some of the people that were involved in the necropsy of dead harbor seals (*Phoca vitulina*) during an IAV epizootic on Cape Cod, USA, in 1979, developed conjunctivitis [26].

The genus *Morbillivirus*, within the family *Paramyxoviridae*, is comprised of enveloped viruses that contain a single-stranded, non-segmented RNA genome. There are seven species that are currently recognized within this genus [27], three of which have been linked to disease in marine mammals: canine distemper virus (CDV; domestic dog and wild carnivore species), phocine distemper virus (PDV; pinnipeds), and cetacean MV (CeMV; cetaceans) [28]. CeMV includes three characterized strains: the dolphin MV (DMV) [29,30], the porpoise MV (PMV) [31], and the pilot whale MV (PWMV) [32]. Novel strains have also been detected by RT-PCR in other cetacean species [33,34,35]. A systemic disease that results in variable degrees of morbidity and mortality characterizes infections with MVs [36]. Abs against MV have been detected in white whales in Russia (Table 1). The only other Arctic cetacean that has been investigated is the narwhal from Canada, which had no MV Abs [37].

The genus *Brucella* contains an increasing number of species of potentially zoonotic bacteria [38]. *Brucella* spp. were isolated for the first time from marine mammals in 1994 [39] and *Brucella pinnipedialis* (pinnipeds) and *Brucella ceti* (cetaceans) were described in 2007 [40]. Marine mammal brucellae have since been isolated from, and serologically indicated in, numerous pinniped and cetacean species. *B. ceti*-associated pathology is well documented in cetaceans [41]. *Brucella* spp. have been serologically indicated, detected by PCR and isolated from white whales (Table 1). Seropositive fin whales (*Balaenoptera physalus*), minke whales (*Balaenoptera acutorostrata*), sei whales (*Balaenoptera borealis*) [42,43], and narwhals [44] have also been detected in Arctic waters, and *B. ceti* has been isolated from a minke whale [42]. Marine brucellae have been shown to have zoonotic potential, however, there was no known contact with marine mammals for any of the three human cases [45,46,47] and characterization of the isolates showed that they belonged to an unusual sequence type (ST), ST 27 [48].

*Toxoplasma gondii*, a food- and water-borne protozoan parasite, can infect a wide range of species, including domestic and wild animals and humans [49]. Symptomatic toxoplasmosis has been reported from a range of different marine mammal species [50]. Sero-, immunohistochemistry-, and PCR-positive white whales have been reported. However, no seropositive animals were detected among white whales from the U.S. Navy Marine Mammal Program in San Diego, USA, or white whales from Svalbard, Norway (Table 1). The parasite has been shown to induce pathology and disease in white whales [51,52], however, *T. gondii* DNA has also been detected in white whales with no histopathological indication of infection [53]. *T. gondii* Abs have been detected in bowhead whales [54] in Arctic waters, while samples from harbor porpoises (*Phocoena phocoena*) [55], minke whales [56], and narwhals [57] have been seronegative. A *T. gondii* serosurvey of Inuit adults from Nunavik, Quebec, Canada, showed a seroprevalence of almost 60%; risk factors for seropositivity included consumption of seal meat [58]. Zoonotic infections with *T. gondii* from marine mammals have been associated with consumption of meat that has not been heated sufficiently to inactivate the parasite [59]. 

Our knowledge about the presence of pathogens in Svalbard white whales is limited. The aim of this study was to investigate Svalbard white whales for evidence of exposure to IAV, MV, *Brucella* spp., and *T. gondii*, which are all potential cetacean pathogens. Knowledge regarding their presence and prevalence in one of the smallest populations of white whales in the world is important, not only to evaluate potential population threats from climate change-induced alterations of the physical environment and the marine ecosystem, but as baseline data for future reference and long-term monitoring. 

**Table 1 pathogens-12-00058-t001:** An overview of previous investigations on the exposure of white whales (*Delphinapterus leucas*) to influenza A virus, morbillivirus, *Brucella,* and *Toxoplasma gondii*. Presented as prevalence (%), number of animals investigated and origin.

Pathogen	Antigen (Isolation/PCR)	Antibodies (Serology)
Influenza A virus	0%, *n* = 1, Ca ^1^ [25]	1%, *n* = 418, Ca [25]
Morbillivirus		0%, *n* = 445, Ca [37]
		14%, *n* = 147, Ru ^2^ [60]
		23%, *n* = 78, Ru [61]
*Brucella* spp.	100%, *n* = 3, Ca [62]	6%, *n* = 488, Ca [44]
	14%, *n* = 69, USA ^3^ [63]	61%, *n* = 167, USA [63]
		7%, *n* = 147, Ru [60]
		42%, *n* = 78, Ru [61]
		75%, *n* = 4, Ru [64]
*Toxoplasma gondii*	1%, *n* = 472, Ca [51,52,65,66]	27%, *n* = 22, Ca [51]
	13%, *n* = 23, Ca [53]	22%, *n* = 46, Ca [54]
	44%, *n* = 34, Ca [67]	5%, *n* = 147, Ru [60]
		14%, *n* = 78, Ru [61]
		0%, *n* = 3, Cp ^4^ [50]
		0%, *n* = 12, Sv ^5^ [57]

^1^ Canada = Ca, ^2^ Russia = Ru, ^3^ The United States of America = USA, ^4^ in captivity = Cp, ^5^ Svalbard Archipelago (Norway) = Sv.

## 2. Materials and Methods

### 2.1. Sampling of White Whales

White whales (*n* = 27) were live captured in 2001–2006 at various locations in the Svalbard Archipelago (Table 2, Figure 1); for details see [3,68]. Briefly, the whales were caught in nets that were set from the shore and physically restrained for a short period in shallow water using a rope around the caudal peduncle and a hoop-net held over the head. Blood was collected from all 27 white whales from the caudal vein into vacuum tubes (without anticoagulants). The blood was left to clot for 4–6 h before serum was obtained by centrifugation (4000 rpm, 10 min, RT°). The serum samples were subsequently kept at −20 °C until analysis. The material included samples from one sub-adult female and 26 adults (22 males and 4 females). Age was determined based on body size and skin color [69] and sex was genetically determined by the DNA-laboratory at Bioforsk Svanhovd (Svanvik, Norway). The standard length of all animals was measured in a straight line centrally on the dorsal side from the tip of the head to the notch in the fluke (Table 2). The Norwegian Food Safety Authority research program approval number was 5158.

### 2.2. Extraction of RNA

Nucleic acid was extracted from serum samples (*n* = 27) using the KingFisher Duo Prime Purification System (ThermoFisher Scientific, Waltham, MA, USA) and the IndiMag Pathogen Kit (Indical Bioscience, Leipzig, Germany). To exclude false-negative results caused by PCR inhibitors and to verify RNA extraction and amplification, an RNA template control (intype IC-RNA, Indical Bioscience) was added to each sample. Due to negative internal controls, the RT-PCRs could not be evaluated for serum samples from two adult males, one captured in Wichebukta (9; DL 99/10; 2001) and the other in Tempelfjorden (27; DL 16/05; 2016) (Table 2).

### 2.3. Influenza A Virus

All 27 serum samples were screened for IAV Abs by an enzyme-linked immunosorbent assay (ELISA: ID Screen influenza A Antibody Competition Multi-species; ID vet, Grabels, France) in 1:2 dilution according to the manufacturer’s instructions. Samples were run in single wells. Controls that were provided by the manufacturer were used in duplicate and the mean value was used to calculate the blocking percent. To obtain the blocking percent of a serum, the optical density (OD) at 450 nm of a sample was divided with the mean of the negative controls (S/N) and expressed in percent.

A total of twenty-five serum samples were investigated for IAV RNA by a one-step Taq-Man qRT-PCR performed on a CFX96 Touch Real-Time PCR Detection System (Bio-Rad Laboratories, Feldkirchen, Germany) using the QuantiTect Probe RT-PCR Kit (Qiagen, Hilden, Germany). The primers (NP-1448-F: 5′-GGG AGT CTT CGA GCT CTC-3′ and NP-1543-R: 5′-GCA TTG TCT CCG AAG AAA TAA GA-3′) and probe (AIV-NP-1473: 5′-Fam-AAG GCA VCG ARC CCG ATC GTG C-Tamra-3′) targeted the nucleoprotein (NP) (WOAH, FAO and National Reference Laboratory for Avian Influenza, Friedrich-Loeffler-Institut, Greifswald-Riems, Germany). Each reaction was set up in a total volume of 20 µL and contained 5 μL of sample RNA as template. The following temperature profile was applied: 50 °C, 30 min; 95 °C, 15 min; 40 × (95 °C, 30 s; 56 °C, 30 s; 72 °C, 30 s). Positive and negative RNA controls were included in each run.

### 2.4. Morbillivirus

The virus neutralization test was used to investigate three samples for neutralizing Abs against PDV isolate 2558/Han 88, using 96-well plates with Vero SLAM cell monolayer [70,71]. Serum samples were titrated in duplicate in dilutions (1:10–1:1280) and incubated with virus (100 TCID_50_) for one hour at 37 °C. The dilutions with virus were then inoculated onto the Vero SLAM cell monolayers. The neutralizing capacity of the sera was evaluated after three days. Samples with a complete absence of cytopathic effect were considered positive.

A SYBR green real-time RT-PCR was used to analyze twenty-five samples for the presence of MV RNA. For this, primers MVP 2202 (5′-KKC TCR TGG TWC CWR CAG GC-3′) and MVP 2480 (5′-TCT CTY CTG TGC CCT TTT TAA TGG-3′) [72] and the QuantiTect SYBR^®^ Green RT-PCR Kit (Qiagen) were used following the manufacturer’s instructions. The PCR reaction mix was adjusted to a total volume of 20 µL containing 5 µL of sample RNA. The temperature profile was 50 °C, 30 min; 95 °C, 15 min; 40 × 95 °C, 30 s; 56 °C, 30 s; 72 °C, 30 s. The PCR was followed by melt curve analysis (65–95 °C, 0.5 °C increment). A no-template control, an internal RNA control (see above), and a PDV RNA positive control from cell culture supernatant were included. A sample was deemed negative when there was no signal within 40 cycles.

### 2.5. Brucella *spp.*

All 27 samples were analyzed for *Brucella* Abs with a Protein A/G indirect ELISA (iELISA) as previously described [43]. The mean optical density (OD) of duplicate wells was expressed as a percentage of the reactivity of the positive control: percent positivity (%P) = (OD sample/OD positive control) × 100. The cut-off was based on the mean value of the iELISA %P plus 2.58 standard deviations for sei-, fin-, and minke whale samples that were classified seronegative in the complement fixation test (CFT), the slow agglutination of Wright (SAW), SAW with ethylenediamine tetra-acetic acid (SAW-EDTA), the Rose Bengal test (RBT), a protein G iELISA and an anti-complement ELISA. This provided a cut-off of 30.8%P for the iELISA [43]. 

### 2.6. Toxoplasma gondii

The ID Screen^®^ Toxoplasmosis Indirect Multiple-Species ELISA kit (ID.vet Innovative Diagnostics) was used to test for the presence of *T. gondii* Abs in serum samples from 27 white whales following the manufacturer’s instructions and controls. The results are expressed as S/P% = (OD sample–OD negative control)/(OD positive control–OD negative control). All samples were run in duplicate.

## 3. Results

Two of the samples (2; DL 97/06 and 14; DL 13/02) showed reactivity in the IAV Ab ELISA with OD values at 450 nm of 0.37 and 0.39. The percent ratios of sample to negative control (S/N%) were calculated to be 43.0% (positive, cut-off: <45%) and 45.3% (doubtful, cut-off: 45–50%), respectively. One sample (21; DL15/01) showed 50.0% blocking activity (OD at 450 nm value of 0.43) being on the cut-off between doubtful and negative results and was classified as negative. The serum sample (2; DL 97/06) that had a clear positive reaction in the competition ELISA originated from a 450 cm long adult male that was captured in Van Mijenfjorden in 2001 (*n* = 1/27, 4%, Figure 1). The serum (14; DL 13/02) that had a doubtful positive result came from a 400 cm long adult male captured in Tempelfjorden in 2013. IAV RNA was not detected in any of the serum samples (*n* = 0/25, 0%, Table 2, Appendix A). 

Due to shortage of serum, only three white whales were investigated for PDV neutralizing Abs, all were classified as seronegative (*n* = 0/3, 0%). MV RNA was not detected in any of the 25 investigated serum samples (*n* = 0/25, 0%, Table 2, Appendix A). 

*Brucella* spp. Abs were detected in 16 serum samples (*n* = 16/27, 59%); 12 adult males, 1 subadult female, and 3 adult females (Table 2, Appendix A).

All 27 white whales were negative for *T. gondii* Abs (*n* = 0/27, 0%, Table 2, Appendix A).

## 4. Discussion

This is the first investigation of Svalbard white whales for evidence of exposure to IAV, MV and *Brucella* spp. and the second investigation for *T. gondii* [57]. The results confirm that Svalbard white whales have been exposed to IAV and *Brucella* spp. There was no evidence of exposure to MV or *T. gondi.* The methods that were used herein have been used previously and have been validated for the detection of IAV Abs [73], IAV-specific RNA [16,24], MV Abs [74,75], MV-specific RNA [72], *Brucella* Abs [43], and *T. gondii* Abs [55]. We are, therefore, confident that the results reflect the exposure status of the white whales investigated accurately.

Abs against IAV were detected in a serum sample from a white whale that was captured in August 2001 (Table 2). Studies on Ab kinetics in animals following natural infections are limited [76], especially in wildlife [77]. In humans, Ab levels peak at 4–7 weeks post-infection after natural infection with IAV H1N1 [78]. In pigs (*Sus scrofa*) that were experimentally infected with IAV (H1N1), Abs can be detected by day 7 post-infection [79]. Multiple studies have indicated that the Ab responses following natural AIV infections can be very long-lived (humans—decades; [80], pigs—28 months; [76], and wild raccoons (*Procyon lotor*)—9 months; [77]). The time of exposure to IAV for the seropositive white whale is, therefore, difficult to estimate and could range from weeks to years prior to sampling.

All the different subtypes of IAV that have previously been detected in cetaceans show strong evidence for having been derived from avian sources [15]. Infected birds can shed avian IAV in their saliva, nasal secretions, and feces [81]. Contact between marine mammals and wild birds at hauling-out sites or when feeding on the same marine food resources, can facilitate cross-species transmission. Waterborne transmission [21] or transmission to whales preying on seabirds has also been suggested as a possible infection route [25]. The transmission from birds to marine mammals is thought to help IAVs acquire further mutations that allow for more efficient replication in mammalian hosts [16]. There have been several outbreaks involving various subtypes of avian IAV in Europe, Canada, the USA, South America, Asia, and Australia [82] during the five years prior to the detection of the seropositive Svalbard white whale in 2001. The first detection of IAV in a bird in Svalbard was in June 2022 when the virus was found in a glaucous gull (*Larus hyperboreus*) in Longyearbyen [83]. However, the presence of the virus in white whales in 2001, as well as in birds in other parts of the Arctic [81], suggests that it was present in birds in Svalbard before its discovery in the avian community in 2022. A bird carrying the virus from lower latitudes during the northbound spring migration is the most likely source of infection for the seropositive white whale that was detected in our study. 

Detecting IAV-specific RNA in blood or serum could indicate a viraemic state, but IAV RNA was not detected in any of the investigated serum samples. IAV viremia has been reported on a few occasions during acute-phase illness in humans [84,85,86]. In mice that were infected with IAV intravenously, viremia persisted for approximately a week [87]. If viremia in a white whale has the same short duration, the probability of detecting IAV RNA in the blood is small. 

In humans, viremia is observed mainly in patients that are infected with IAV subtypes that have high pathogenic potential, such as H5N1 [84,85,86] and is also associated with a poor prognosis [85,86,88,89]. Previously, the subtypes H1N3 [18], H13N2, and H13N9 [21] have been isolated from lung, liver, and hilar lymph nodes in cetaceans. The two latter, H13N2 and H13N9, viral subtypes seem to have pathogenic potential in whales, as they have been isolated from animals involved in two long-finned pilot whale stranding events along the New England coast in October and November 1984 [21]. The pathological changes included large hilar lymph nodes and hemorrhagic lungs [21,90]. We do not know which IAV subtype the seropositive white whale was exposed to, however, infection with the more pathogenic subtypes could lead to morbidity and mortality in Svalbard white whales.

Blowhole swabs are a simple and non-invasive method for collecting samples from cetaceans, which is suited to investigate the presence of respiratory pathogens [91]. The relative sensitivity and specificity of real-time PCR for *Brucella* spp. on blowhole swabs from bottlenose dolphins (*Tursiops truncatus*) as compared to the real-time PCR on lung samples was 94% (17/18) and 100% (63/63), respectively [92]. Including blowhole swabs in the present study might have yielded relevant results concerning bacterial and viral pathogens that were present in the respiratory system of white whales. 

The three serum samples that were investigated for MV Abs were classified as being negative, however, due to the low number of samples investigated it is difficult to conclude whether MV Abs are truly absent in this population. Abs against MV have previously been detected in white whales in Sakhalinsky Bay, Russia, with a seroprevalences up to 23% [61] showing that white whales have the capacity to harbor MV infection (Table 1). MV RNA was not detected in the white whale serum samples that were investigated in our study. This is in line with the serological results and could indicate that the animals were not infected, however, these findings could also reflect that viremia in MV-infected white whales could be short-lived (weeks), as has been observed in dogs (*Canis lupus familiaris*) and ferrets (*Mustela putorius furo*) that are infected with CDV [93].

Infections with MVs in marine mammals are characterized by generalized immunosuppression, systemic disease, bacterial or viral co-infections, and variable degrees of morbidity and mortality [36]. Over the last three decades, CeMVs have been responsible for several large disease outbreaks among cetaceans. However, some cetacean species and populations have low mortality rates related to enzootic MVs. These populations may function as reservoirs and vectors of the infection to other susceptible species and populations. MVs are extremely infectious, spreading mostly by horizontal dissemination of aerosolized virus, or from mother to calf via the placenta or during lactation, and are likely to infect most immunologically naive individuals in a population [28]. The MV status of the Svalbard white whale population is still unknown. 

*Brucella* spp. Abs were detected in 59% of the white whale serum samples that were investigated in our study. This is in accordance with a recent study involving 167 white whales from Bristol Bay and the eastern Chukchi Sea, Alaska, USA, where an overall seroprevalence of 63% was found [63]. It is, therefore, not very surprising that white whales in Svalbard are exposed to *Brucella* spp. Neurological symptoms such as opisthotonus, tremors, seizures, disorientation, and an inability to maintain buoyancy have been associated with *B. ceti* infection of the central nervous system in multiple cetacean species. The pathologic changes include spinal discospondylitis, meningoencephalitis, meningitis, the eye disorder choroiditis, altered cerebrospinal fluid, and remodeling of the occipital condyles. *B. ceti* has also been associated with mastitis, placentitis, endometritis, genital ulcers, abortion, and testicular abscess in cetaceans [41,94]. It is unknown whether white whales have pathological changes that are similar to those found in other cetacean species that are infected with *Brucella* spp., however a high prevalence of seropositivity (59%) combined with a low level of PCR (14%) and culture (0%) positive results in Alaskan white whales suggested widespread exposure, but a lack of clinical disease [63]. The three *Brucella*-isolates that have been retrieved from white whales belonged to a distinct *B. pinnipedialis* ST (ST25) that is normally associated with true seals and sea otters (*Enhydra lutris*) [62]. Whether this implies a lower pathogenicity in white whales, as observed in true seals infected with *B. pinnipedialis* [41], is unknown, but it is possible that *Brucella* spp. infections in white whales have restricted pathogenic potential.

The negative results for *T. gondii* Abs in the present study were in accordance with a previous investigation of Svalbard white whales [57]. The parasite is, however, present in other wild animals in Svalbard, such as ringed seals (*Pusa hispida*), bearded seals (*Erignathus barbatus*), polar bears (*Ursus maritimus*), arctic foxes (*Vulpes lagopus*), and several bird species [57,95,96,97]. Typically, domestic cats and wild felids are the main hosts, which shed oocysts of *T. gondii* in their feces, which contaminates the environment, food, and water. Intermediate hosts are infected after consuming oocysts, which develop into tissue cysts. In turn, consumption of these infected tissues leads to infection in felid definitive hosts as well as carnivorous intermediate hosts [49]. However, no wild felids are present on Svalbard, and domestic cats are prohibited, so if there are cats in settlements, the population is likely very small. The *T. gondii* life cycle on Svalbard is unknown but the absence of Abs against *T. gondii* in Svalbard reindeer (*Rangifer tarandus platyrhynchus*) and sibling voles (*Microtus rossiaemeridionalis*) indicates that transmission of the parasite by oocysts is not likely an important mechanism in the Svalbard ecosystem. Furthermore, the relatively high seroprevalence among arctic foxes and the presence of Abs in barnacle geese (*Branta leucopsis*), suggest that migratory birds are the most likely vectors bringing the parasite to Svalbard [95].

Macroscopic pathology in white whales that are infected with *T. gondii* include enlargement of all lymph nodes, with a pale and wet section surface, and petechiae at the junction of the grey and white matter of the brain [51]. In addition, unilateral sclerosing mastitis and renal hemorrhages have been described [51,52]. Histopathological lesions have included lymph node medullary sinuses enlarged with macrophages and parasite larval stages (tachyzoites). Moderate histiocytic infiltration and marked diffuse lymphoid depletion in the thymus and in the lymphoid tissue of the anal mucosa has been associated with *T. gondii*. Tissue cysts and tachyzoites have also been observed in the spleen and lungs [51,52]. *T. gondii* DNA has also been detected in white whales with no histopathological indication of infection [53]. Thus, we know that white whales are susceptible to *T. gondii* infection and that the parasite may cause pathology and disease but as of yet, no seropositive individuals have been identified in the Svalbard population. Accumulated evidence shows that changing environmental factors can exert influence on the occurrence, transmission, and distribution of *T. gondii* [98]. Whether the Svalbard white whale population remains *T. gondii* negative with the ongoing climate changes warrants further investigations and continued surveillance.

Climate change may affect disease prevalence through a wide range of direct and indirect mechanisms, including altered animal and pathogen species ranges, weakened immune response due to stress, toxicant exposures, as well as habitat degradation (loss of sea ice) and effects on body condition due to shifts in the food web [99]. In addition, factors that are associated with increased human habitation and traffic in the Arctic may play a role in disease transmissions and effects [10,12]. These alterations may contribute to altered pathogen exposure and disease occurrence for the Svalbard white whale population. 

## 5. Conclusions

IAV Abs were detected in an adult male white whale that was captured in 2001, although no IAV RNA was detected. *Brucella* spp. Abs were found in 59% of the investigated white whales while all MV and *T. gondii* results were negative.

There is substantial evidence that diseases can impact wildlife populations by causing temporary or permanent declines in abundance. The Svalbard white whale population is one of the smallest white whale populations in the world, making it vulnerable to challenges. Dramatic changes in climate and marine ecosystems are also taking place in the Arctic, which may affect disease prevalence through a range of mechanisms. 

Detection alone, as presented herein, is therefore insufficient to evaluate the health status of the Svalbard white whale population. A continuous surveillance of health parameters, including pathogen exposure, is critical for monitoring the status of this vulnerable population of white whales. 

## Figures and Tables

**Figure 1 pathogens-12-00058-f001:**
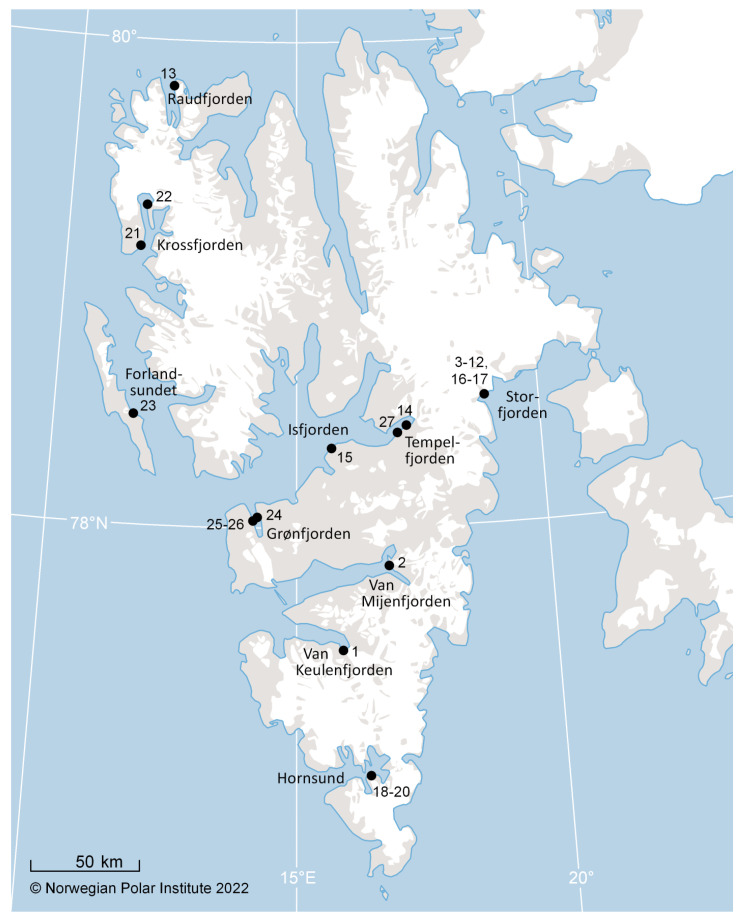
Sampling locations for Svalbard white whales (*Delphinapterus leucas*). Map of the Svalbard Archipelago showing the sampling locations for white whales (*Delphinapterus leucas*) that were investigated for evidence of exposure to influenza A virus, morbillivirus, *Brucella* spp., and *Toxoplasma gondii*. The numbers on the map correspond with the numbers used in Table 2.

**Table 2 pathogens-12-00058-t002:** An overview of Svalbard white whales (*Delphinapterus leucas*) that were included in the study and results from serological tests (Ab) and detection of pathogen-specific RNA. The numbers given in the first column correspond with the numbers provided in Figure 1.

No	ID	Yr	Location	Sex	Length ^1^	Age Cat.	IAV ^2^ Ab	IAV RNA	PDV ^3^ Ab	MV ^4^ RNA	*Tg*^5^ Ab	*Bru*^6^ Ab
1	DL 96/01	01	Van Keulenfjorden	M ^7^	435	A ^9^	0	0	-	0	0	0
2	DL 97/06	01	Van Mijenfjorden	M	450	A	1	0	-	0	0	1
3	DL 98/06	01	Storfjorden	M	370	A	0	0	-	0	0	1
4	DL 99/02	01	Storfjorden	M	425	A	0	0	-	0	0	0
5	DL 99/04	01	Storfjorden	M	405	A	0	0	0	0	0	1
6	DL 99/05	01	Storfjorden	F ^8^	260	Sa ^10^	0	0	-	0	0	1
7	DL 99/06	01	Storfjorden	M	430	A	0	0	-	0	0	0
8	DL 99/09	01	Storfjorden	M	465	A	0	0	-	0	0	0
9	DL 99/10	01	Storfjorden	M	440	A	0	-	-	-	0	0
10	DL 99/11	01	Storfjorden	M	410	A	0	0	-	0	0	1
11	DL 99/12	01	Storfjorden	F	370	A	0	0	-	0	0	1
12	DL 99/15	01	Storfjorden	F	380	A	0	0	-	0	0	1
13	DL 13/01	13	Raudfjorden	M	440	A	0	0	-	0	0	0
14	DL 13/02	13	Tempelfjorden	M	400	A	Do ^11^	0	-	0	0	1
15	DL 13/03	13	Isfjorden	M	455	A	0	0	0	0	0	1
16	DL 14/02	14	Storfjorden	F	410	A	0	0	-	0	0	0
17	DL 14/03	14	Storfjorden	M	405	A	0	0	-	0	0	1
18	DL 14/04	14	Hornsund	M	395	A	0	0	-	0	0	1
19	DL 14/06	14	Hornsund	F	375	A	0	0	-	0	0	1
20	DL 14/08	14	Hornsund	M	380	A	0	0	-	0	0	1
21	DL 15/01	15	Krossfjorden	M	440	A	0	0	0	0	0	0
22	DL 15/02	15	Krossfjorden	M	340	A	0	0	-	0	0	0
23	DL 16/01	16	Forlandsundet	M	435	A	0	0	-	0	0	0
24	DL 16/02	16	Grønfjorden	M	430	A	0	0	-	0	0	1
25	DL 16/03	16	Grønfjorden	M	365	A	0	0	-	0	0	1
26	DL 16/04	16	Grønfjorden	M	385	A	0	0	-	0	0	1
27	DL 16/05	16	Tempelfjorden	M	425	A	0	-	-	-	0	0

^1^ The standard length was measured in a straight line centrally on the dorsal side from the tip of the head to the notch in the fluke. ^2^ influenza A virus = IAV. ^3^ Phocine distemper virus = PDV. ^4^ Morbillivirus = MV. ^5^ *Toxoplasma gondii* = *Tg*. ^6^
*Brucella* = *Bru.*
^7^ Male = M. ^8^ Female = F. ^9^ Adult = A. ^10^ Subadult = Sa. ^11^ Doubtful result = Do.

## Data Availability

Not applicable.

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
