# Peer review of "Pathogen Exposure in White Whales (Delphinapterus leucas) in Svalbard, Norway"

_pathogens, 2022, doi:10.3390/pathogens12010058_

Round 1
Reviewer 1 Report
The aim of this study was to investigate Svalbard white whales for evidence of exposure to IAV, MV, Brucella spp., and T. gondii, which are all potential cetacean pathogens. Knowledge regarding their presence and prevalence in one of the smallest populations of white whales in the world is important, not only to evaluate potential population threats from a changing climate and living conditions but as baseline data for future reference and long-term monitoring. The manuscript is very interesting and covers an important topic. Additionally, it is well-written and its presentation is very good. However, I have some major remarks before performing a detailed evaluation of this manuscript. The comments are detailed above:
1 – I suggest to authors divide their Materials and methods into subsections: Sampling, isolation, culture, …….etc. What was the number of samples collected?
2- This results in close contact with distinct species would be very interesting to analyze. The paper is already exciting but authors can obtain more useful information with their data.
3– I think the results data are very low in the manuscript, why the authors attach a Supplementary File. I think the results Supplementary File can summarize and include in the main text.
4- Abstract needs to increase
5- How was the sampling scheme and sample calculated? Was it representative? This information is very important to understand the representativeness of this study.
6- Were samples obtained from more than one individual that works together? This must be useful to understand the spa typing.
7 – The discussion is very comparative and doesn’t discuss possible reasons for antimicrobial resistance in the area.
8 - How the One Health concept can be useful to avoid this problem in Norway?
9- Please all scientific names.
Reviewer 2 Report
Thank you for the opportunity to review this paper.
Overall a well written paper needing few editorial changes. I did have a few comments for improvement and some questions about methods. First, there is an extensive review of the different pathogens in the introduction which seemed a bit repeated in the discussion. Perhaps some info from the introduction could move to the discussion? I also wondered about using serum for the PCR for the viral agents. This would be highly unusual to detect these agents in the serum and I wonder why Blow hole swabs were not collected from live capture animals. If you are trying to detect a current infection with respiratory pathogens such as influenza and DMV, BH swabs would be a better option. Perhaps that was not possible with the capture circumstances? If not, why – if you can get blood, you should be able to get swabs and there are ways to preserve these samples without extreme storage requirements (RNALater). Perhaps a discussion about this would be helpful perhaps after your comments on the low likelihood of detecting virus in the serum (line 318). Otherwise there are minor comments by line:
Line 103 “including” not included
Table 1: there is an extra space between Antigen and (isolation / PCR)
Table 2: You need to define the following in the legend: M, A. Not done vs No result is more commonly used for samples not tested.
Round 2
Reviewer 1 Report
Accept in present form